A retrospective study investigating the clinical significance of body mass index in acute pancreatitis

Bai Yuanzhen
Gong Guanwen
Aierken Reziya
Liu Xingyu
Cheng Wei
Guan Junjie
Jiang Zhiwei fsyy00630@njucm.edu.cn
Jiangsu Province Hospital of Chinese Medicine, Nanjing University of Chinese Medicine , Nanjing , China
Oliveira Sonia
Electronic publication date: 2024 Jan 29
Publication date: 2024
Volume: 12
Electronic Location ID: e16854
Received 2023 Aug 23; Accepted 2024 Jan 8
Copyright: © 2024 Bai et al.
Copyright year: 2024
Copyright holder: Bai et al.
License: This is an open access article distributed under the terms of the Creative Commons Attribution License, which permits unrestricted use, distribution, reproduction and adaptation in any medium and for any purpose provided that it is properly attributed. For attribution, the original author(s), title, publication source (PeerJ) and either DOI or URL of the article must be cited.
License URL: https://creativecommons.org/licenses/by/4.0/

Keywords: Body mass index, Acute pancreatitis, Obesity, Severity, Predictor

Funding: National Natural Science Foundation of China 81704083 Special research project of Jiangsu Provincial Administration of Traditional Chinese Medicine ZT202108 This work was supported by the National Natural Science Foundation of China (grant numbers 81704083) and the Special research project of Jiangsu Provincial Administration of Traditional Chinese Medicine (grant number ZT202108). The funders had no role in study design, data collection and analysis, decision to publish, or preparation of the manuscript.

==============================
Background

Acute pancreatitis is an unpredictable and potentially fatal condition for which no definitive cure is currently available. Our research focused on exploring the connection between body mass index, a frequently overlooked risk factor, and both the onset and progression of acute pancreatitis.

Material/Methods

A total of 247 patients with acute pancreatitis admitted to Jiangsu Provincial Hospital of Chinese Medicine from January 2021 to February 2023 were retrospectively reviewed. After screening, 117 patients with complete height and body weight data were selected for detailed assessment. Additionally, 85 individuals who underwent physical examinations at our hospital during this period were compiled to create a control group. The study received ethical approval from the ethics committee of Jiangsu Province Hospital of Chinese Medicine (Ref: No.2022NL-114-02) and was conducted in accordance with the China Good Clinical Practice in Research guidelines.

Results

A significant difference in body mass index (BMI) was observed between the healthy group and acute pancreatitis (AP) patients (p < 0.05), with a more pronounced disparity noted in cases of hyperlipidemic acute pancreatitis (p < 0.01). A potential risk for AP was identified at a BMI greater than 23.56 kg/m2 (AUC = 0.6086, p < 0.05). Being in the obese stage I (95%CI, [1.11–1.84]) or having a BMI below 25.4 kg/m2 (95%CI, [1.82–6.48]) are identified as risk factors for adverse AP progression. Moreover, BMI effectively predicts the onset of acute edematous pancreatitis and acute necrotizing pancreatitis (AUC = 0.7893, p < 0.001, cut-off value = 25.88 kg/m2). A higher BMI correlates with increased recurrence rates within a short timeframe (r = 0.7532, p < 0.01).

Conclusions

Elevated BMI is a risk factor for both the occurrence and progression of AP, and underweight status may similarly contribute to poor disease outcomes. BMI is crucial for risk prediction and stratification in AP and warrants ongoing monitoring and consideration.

Background

A survey of 19.2 million participants in 200 countries has revealed a major shift over the past four decades: the prevalence of underweight individuals was once more than double that of obesity, but now there are more obese than underweight people (NCD Risk Factor Collaboration (NCD-RisC), 2016). This transition highlights the critical health crisis of obesity that needs global attention. Unfortunately, current interventions and policies have not successfully halted the rise in body mass index (BMI) in most countries (Ezzati & Riboli, 2012; Kleinert & Horton, 2015; Roberto et al., 2015). BMI, a standard measure of health based on weight and height, indicates whether a person is under or overweight. High BMI, a known risk factor for hypertension, diabetes mellitus, digestive system cancers, and musculoskeletal disorders, has been confirmed in various studies (Drøyvold et al., 2005; Wei et al., 2019; Larsson, Spyrou & Mantzoros, 2022; Zhao et al., 2022).

Pancreatitis is one of the leading causes for GI-disease related hospital admissions and it is associated with considerable morbidity, mortality and socioeconomic burden (Peery et al., 2019). Acute pancreatitis (AP) can be characterised as acute episodes of persistent severe epigastric pain, accompanied by abdominal distension, nausea and vomiting. Most (~80%) AP cases are mild, resulting in minimal damage to the pancreas and with a relatively short recovery period (Banks et al., 2013). However, around 10–20% of acute pancreatitis patients progress to severe acute pancreatitis (Zhao et al., 2022). This severe form is marked by an intense inflammatory response in its early phase and the development of infected pancreatic necrosis in the later phase (Al Mofleh, 2008). This raises an important question: why do some patients with acute pancreatitis (AP) develop complications like multiple organ dysfunction syndrome, sepsis, or even face mortality, whereas others do not? Identified risk factors, such as age, triglyceride levels, the degree of localized pancreatic damage, and genetic factors, have been shown to predict the onset of organ failure in AP patients (Garg & Singh, 2019). However, the role of obesity as a risk factor for adverse outcomes in AP remains under-analyzed. Evidence from population-based cohort studies has demonstrated a clear association between being overweight or obese and the development of severe acute pancreatitis (SAP) (Dobszai et al., 2019). Conversely, other studies have arrived at contrary conclusions (Párniczky et al., 2016). Choi et al. (2022) posited that a low BMI might also elevate AP risk in individuals with type two diabetes.

The aim of this research is to clarify the causal link between the incidence and progression of acute pancreatitis and BMI, explore the correlation between BMI categories and the severity of acute pancreatitis, and thereby offer additional insights into focused prevention and treatment approaches for this condition.

Materials and Methods

Study design

Between January 10, 2021, and February 10, 2023, Jiangsu Provincial Hospital of Chinese Medicine in Jiangsu, China, treated 247 acute pancreatitis patients. In this retrospective analysis, 117 patients with detailed height and body weight records were selected, consisting of 65 males and 52 females. Comprehensive patient information, including age, gender, height, weight at admission, triggers, smoking and drinking habits, menopausal status, history of abdominal surgery, initial episodes or recurrences, recurrences within 6 months post-discharge, etiologies, hospital stay duration, and presence of diabetes mellitus, hypertension, or steatohepatitis was meticulously recorded. Additionally, 85 individuals undergoing physical examinations at our hospital during the same timeframe were included as a control group. This group had no history of acute pancreatitis but were not excluded for smoking, alcohol consumption, abdominal surgery, or conditions like diabetes mellitus, hypertension, or steatohepatitis. There were no significant differences in gender and age between the control and study groups. Data compilation for this research occurred from January 27 to February 21, 2023. Ethical approval was secured from the ethics committee of Jiangsu Province Hospital of Chinese Medicine (Ref: No.2022NL-114-02), and the study adhered to the updated Good Clinical Practice-2020 guidelines in China (Yang et al., 2021). The study was also in alignment with the Declaration of Helsinki. Written informed consent was obtained from all participants involved in the study.

Inclusion and exclusion criteria

The study included patients aged 20 to 75 years who fulfilled the diagnostic criteria for acute pancreatitis (AP) (Boxhoorn et al., 2020) and were admitted within 3 days of pain onset. Exclusion criteria encompassed patients who declined treatment, those with severe psychiatric disorders, chronic pancreatitis, pancreatic cancer, a history of pancreatic surgery, pregnancy, incomplete data, or contraindications to contrast agents.

Evaluation criteria

Definition of severity of acute pancreatitis

The revised Atlanta classification (RAC) defines three degrees of severity (Banks et al., 2013): (1) mild acute pancreatitis (MAP): no local or systemic complications and organ failure, (2) moderately severe acute pancreatitis (MSAP): local or systemic complications, transient organ failure (<48 h), or both (local complications include peripancreatic fluid collections and acute necrotic collections), (3) severe acute pancreatitis (SAP): local or systemic complications, and persistent single or multiple organ failure (>48 h) (the diagnostic criteria for organ failure were based on the modified Marshall scoring system (Foster et al., 2019)). The severity of AP was evaluated within 48 h of admission.

Definition of types of acute pancreatitis

Acute pancreatitis can be subdivided into two types: edematous pancreatitis (AEP) and necrotising pancreatitis (ANP). The former refers to diffuse (or occasionally localised) enlargement of the pancreas due to inflammatory oedema. On contrast-enhanced computed tomography (CECT), the pancreatic parenchyma shows relatively homogeneous enhancement, and the peripancreatic fat usually shows some inflammatory changes of haziness or mild stranding. The latter showed necrosis of the pancreatic parenchyma, the peripancreatic tissue or both (Banks et al., 2013). In this study, CECT examination of AP patients was performed 72 h after admission to determine the type.

Body mass index

BMI was calculated as measured weight in kilograms divided by measured height squared (kg/m2) at the examination. Through data calculation, BMI was classified into five groups defined by the World Health Organization (WHO) recommendations for Asians (World Health Organization, 2000): underweight (BMI < 18.5 kg/m2), normal-weight (BMI 18.5–22.9 kg/m2), overweight (BMI 23–24.9 kg/m2), obese stage I (BMI 25–29.9 kg/m2), and obese stage II (BMI ≥ 30 kg/m2).

Statistical analysis

Statistical analyses were conducted using GraphPad Prism software, version 9.5.0 (GraphPad Software, La Jolla, CA, USA). A p-value of less than 0.05 was deemed to indicate statistical significance. Depending on whether the distribution was normal, summary statistics were presented as either mean ± standard deviation (SD) or median with interquartile range (IQR). The Shapiro-Wilk test was utilized to assess the normality of numerical variables. One-way analysis of variance (ANOVA) was employed to compare gender, age, and BMI across different groups. The significance of differences in quantitative and qualitative variables was determined using either the Student’s t-test or the Chi-square test. Non-parametric data were evaluated using the Whitney U-test. The impact of various factors on the severity of acute pancreatitis (AP) was analyzed through multivariate logistic regression, with odds ratios (OR) computed alongside 95% confidence intervals (CI). Additionally, the role of BMI in AP was assessed using receiver operating characteristic (ROC) curve analysis, with the area under the ROC curve (AUC) providing a measure of predictive performance.

Results

Patient characteristics

Table 1 presents baseline characteristics segmented by Body Mass Index (BMI). The study enrolled 117 patients, with a median age of 53 years (IQR 40.0–61.0), showing no significant age variation across different BMI categories. Gender differences were not statistically significant, though a higher proportion of males was observed in the obese group, whereas females were more prevalent in the underweight category. The median BMI for the cohort was 25.4 (IQR 23.2–27.5). In terms of etiology, biliary causes were predominant (47.0%), followed by hypertriglyceridemia (24.8%), idiopathic origins (20.5%), and alcohol (7.7%) being the least common. More than half of the female patients in the obese and overweight groups were menopausal. Of all participants, 79 were experiencing their first episode of acute pancreatitis(incipience), while 38 had recurrent episodes (palindromia), with a 10.3% chance of recurrence within 6 months post-discharge. The distribution of pancreatitis severity varied with BMI; both MSAP and SAP were more frequent in obese and overweight groups. Conversely, MAP was mostly seen in underweight and normal-weight groups. Acute edematous pancreatitis was observed across all BMI categories, whereas necrotic pancreatitis was exclusive to the obese group.

Table 1 Baseline characteristics segmented by body mass index (BMI).

	UW (n = 4)	NW (n = 24)	OW (n = 21)	OB I (n = 54)	OB II (n = 14)	All (n = 117)	
Age (years), median (IQR)	61.5 (50.8, 70.8)	57.0 (41.3, 67.5)	46.0 (37.5, 63.0)	50.5 (40.8, 58.3)	50.5 (36.3, 57.8)	53.0 (40.0, 61.0)	
Male, n (%)	1 (25.0)	15 (62.5)	10 (47.6)	30 (54.5)	10 (71.4)	65 (55.6)	
Female, n (%)	3 (75.0)	9 (37.5)	11 (52.4)	25 (45.5)	4 (28.6)	52 (44.4)	
BMI (kg/m2), median (IQR)	16.9 (16.5, 18.0)	21.4 (20.9, 22.2)	23.8 (23.6, 24.4)	26.6 (25.5, 27.8)	32.3 (31.7, 34.0)	25.4 (23.2, 27.5)	
Hospital stay duration (day), medium (IQR)	5.5 (4.3, 6.8)	7.5 (4.0, 9.5)	6.0 (5.0, 8.0)	8.0 (6.0, 10.3)	9.5 (6.8, 11.0)	7.0 (6.0, 10.0)	
Etiology, n (%)							
Biliary	2 (50.0)	11 (45.8)	9 (42.9)	28 (51.8)	5 (35.7)	55 (47.0)	
Alcohol	0	4 (16.7)	1 (4.7)	3 (5.6)	1 (7.1)	9 (7.7)	
Hypertriglyceridemia	1 (25.0)	2 (8.3)	5 (23.8)	15 (27.8)	6 (42.9)	29 (24.8)	
Idiopathic	1 (25.0)	7 (29.2)	6 (28.6)	8 (14.8)	2 (14.3)	24 (20.5)	
Basic disease, n (%)							
Hypertension	1 (25.0)	3 (12.5)	6 (28.6)	18 (33.3)	8 (57.1)	36 (30.8)	
Diabetes mellitus	2 (50.0)	2 (8.3)	2 (9.5)	17 (31.5)	3 (21.4)	26 (22.2)	
Steatohepatitis	0	7 (29.2)	7 (33.3)	21 (38.9)	9 (64.3)	44 (37.6)	
Current smokers, n (%)	1 (25.0)	8 (33.3)	2 (9.5)	8 (14.8)	2 (14.3)	21 (17.9)	
Alcohol intake, n (%)	1 (25.0)	5 (20.8)	1 (4.8)	9 (16.7)	1 (7.1)	17 (14.5)	
Female menopause, n (%)	3 (100.0)	3 (33.3)	7 (63.6)	19 (76.0)	3 (75.0)	35 (67.3)	
History of abdominal surgery, n (%)	2 (50.0)	2 (8.3)	5 (23.8)	28 (51.9)	4 (28.6)	41 (35.0)	
Incipience/Palindromia	4/0	16/8	16/5	32/22	11/3	79/38	
Palindromia within 6 months after discharge, n (%)	0	4 (16.7)	3 (14.3)	5 (9.3)	0	12 (10.3)	
Severity of acute pancreatitis, n (%)							
MAP	4 (100.0)	24 (100.0)	17 (81.0)	31 (57.4)	2 (14.3)	78 (66.7)	
MSAP	0	0	4 (19.0)	16 (29.6)	9 (64.3)	29 (24.8)	
SAP	0	0	0	7 (13.0)	3 (21.4)	10 (8.5)	
Types of acute pancreatitis, n (%)							
AEP	4 (100.0)	24 (100.0)	21 (100.0)	41 (75.9)	10 (71.4)	99 (84.6)	
ANP	0	0	0	13 (24.1)	4 (28.6)	18 (15.4)	
Note:

Categorical variables are percent and continuous variables are median and interquartile range. Abbreviation: UW, underweight (BMI < 18.5 kg/m2), NW, normal weight (BMI 18.5–22.9 kg/m2), OW, overweight (BMI 23–24.9 kg/m2), OB I, obese stage I (BMI 25–29.9 kg/m2) and OB II, obese stage II (BMI ≥ 30 kg/m2). BMI, body mass index; MAP, mild acute pancreatitis; MSAP, moderately severe acute pancreatitis; SAP, severe acute pancreatitis; AEP, acute edematous pancreatitis; ANP, acute necrotising pancreatitis; IQR, inter-quartile range.

Analysis for inducement and etiology

In a cohort of 117 cases with various inducements, the etiology was diverse: improper diet in 45 patients (38.46%), undetermined causes in 35 cases (29.91%), alcohol abuse in 13 (11.11%), fatigue in eight (6.84%), poorly controlled diabetes in seven (5.98%), exposure to cold in five (4.27%), and iatrogenic factors in four (3.42%). The BMI comparison revealed a statistically significant variance between the healthy control group and those with AP (p = 0.042). Among AP cases with different etiologies, the hyperlipidemia subgroup showed a significant BMI disparity compared to the control group (p = 0.003). However, no significant BMI differences were noted among other etiological subgroups of AP, as illustrated in Fig. 1.

Figure 1 Comparative differences in etiologies between healthy controls and AP patients.

The BMI comparison revealed a statistically significant variance between the healthy control group and those with AP (p = 0.042). Significance indicated by *p < 0.05, **p < 0.01.

Predictors of severity

As shown in Fig. 2, the association between a BMI below 25.4 kg/m2 and the severity of MSAP and SAP was more pronounced when stratified by factors such as age, gender, BMI, diabetes mellitus/hypertension/steatohepatitis, smoking and drinking habits, menopausal status, history of abdominal surgery, and palindromia. The odds ratio (95% CI) for BMI < 25.4 kg/m2 was 3.44 (1.82–6.48) and for BMI ≥ 25.4 kg/m2, it was 0.37 (0.25–0.54), both statistically significant (p < 0.05). This indicates that a BMI ≥ 25.4 kg/m2 may act as a protective factor against exacerbations, which contrasts with our initial hypothesis. To delve deeper, given the potential limitations in BMI categorization, we compared severity across different BMI categories: Overweight (OW, 23–24.9 kg/m2), Obesity Class I (OB I, 25–29.9 kg/m2), and Obesity Class II (OB II, ≥30 kg/m2). The severity in the OB II group was markedly distinct from OW and OB I groups (Chi-square, df = 15.25, 2, p < 0.001). Further analysis of OB I and OB II suggests that a BMI between 25 and 29.9 kg/m2 remains a risk factor for SAP progression. While the impact of a BMI ≥ 30 kg/m2 on disease worsening may not be as pronounced as in OW and OB I groups, its potential protective effect warrants further investigation. No cases of moderate to severe pancreatitis were observed in the underweight (UW) or normal weight (NW) groups, potentially due to sample size limitations; hence, no further subgroup analysis was conducted. However, the possibility that a low BMI increases the risk of adverse disease progression cannot be dismissed, aligning with findings from several other studies.

Figure 2 Relationship between predictors and severity of acute pancreatitis.

Multivariate analysis indicating BMI < 25 kg/m2 and OB I as risk factors for AP progression to MSAP and SAP. Significance indicated by **p < 0.01, ***p < 0.001.

Predict the incidence and assist in the diagnosis of acute pancreatitis type

As depicted in Fig. 3, the AUCs for BMI in estimating the risk and aiding the diagnosis of AP types were 0.6086 ± 0.04 and 0.7893 ± 0.04, respectively. The lower AUC for predicting AP (less than 0.7) suggests limited accuracy, with a cut-off value at 23.56 kg/m2 (p < 0.05). However, BMI (AUC = 0.7893) enhances the diagnostic accuracy when combined with CECT in differentiating AEP from ANP, especially at a BMI of ≥25.88 kg/m2 (p < 0.0001). Additionally, while BMI is not a significant predictor for AP recurrence (AUC = 0.5591), a positive correlation exists between BMI and the likelihood of recurrence within 6 months (r = 0.7532, p < 0.01), indicating that higher BMI may be associated with increased recurrence risk.

Figure 3 To evaluate the role of BMI in risk, clinical type, and recurrence of acute pancreatitis.

ROC curve analysis for BMI in predicting AP type incidence and diagnosis assistance, with AUC values of 0.6086 and 0.7893, respectively.

Discussion

This study, involving 117 patients from the Jiangsu Provincial Hospital of Chinese Medicine, reveals several key insights regarding BMI and AP. First, there is a distinct difference in the BMI of AP patients compared to healthy controls, suggesting a potential AP risk at a BMI exceeding 23.56 kg/m2. Second, a significant association was observed between BMI and AP severity. Specifically, OB I or a BMI under 25.4 kg/m2 was identified as a risk factor for developing Mildly Severe/Severe AP (MSAP/SAP). Third, according to ROC analysis, while BMI does not effectively predict AP onset, it is a robust indicator for distinguishing between AEP and ANP, as evidenced by an AUC over 0.7. Finally, our findings indicate a novel observation: the higher the BMI, the greater the likelihood of AP recurrence within a short timeframe (≤6 months).

AP is an inflammatory condition triggered by trypsin activation in the pancreas due to various factors, leading to the digestion, edema, hemorrhage, and potentially necrosis of pancreatic tissue. Studies have shown that triglyceride (TG) levels in hepatocytes of patients with obesity, diabetes, and metabolic syndrome are significantly elevated (Raz et al., 2005). This excessive TG storage, when hydrolyzed by trypsin, releases a large amount of free fatty acids (FFA), directly damaging pancreatic acinar cells and vascular endothelial cells (Stimac et al., 2013). Moreover, the abundance of nutrients and obesity contribute to extensive white adipose tissue remodeling, leading to changes in adipokine production and a low-grade inflammatory response (Balistreri, Caruso & Candore, 2010). Calcium ion overload is another mechanism involved in AP. It promotes the release of lactate dehydrogenase or induces acinar cell apoptosis by activating various cellular zymogens (Gerasimenko, Gerasimenko & Petersen, 2014). The abnormal increase in calcium ions is often due to the excessive unsaturated fatty acids (UFA) produced during TG decomposition, causing the release of calcium ions from the endoplasmic reticulum into the cytoplasm (Petersen et al., 2009). Furthermore, elevated TG can increase blood viscosity and potentially lead to thrombosis, resulting in pancreatic microcirculation disorders and ischemic necrosis. Additionally, dyslipidemia deposits on the inner walls of blood vessels promote the infiltration of inflammatory cells and the release of various inflammatory cytokines (Pooran et al., 2003), playing a pivotal role in the pathogenesis of AP and its systemic complications. This background supports our findings that patients with Obesity Class I (BMI 25–29.9 kg/m2) had more organ failures and local complications than their NW counterparts, confirming the credibility of our results.

The various etiologies of AP are closely associated with obesity. Biliary diseases, such as stones, sludge, or micro-lithiasis in the biliopancreatic passages, can lead to AP by either causing bile reflux or increasing pancreatic duct pressure (Lerch & Gorelick, 2013). Overweight and obese individuals are at a higher risk of developing biliary diseases (Radmard et al., 2015), possibly due to a high-fat diet predisposing them to cholesterol-rich crystals or stones in the bile (Lee, Keane & Pereira, 2015). Additionally, recent epidemiological studies have highlighted an elevated risk of AP in obesity-related metabolic conditions, like type two diabetes, especially when combined with excessive alcohol consumption (Lai et al., 2011). Prior research has shown that acute alcohol administration in obese mice leads to pancreatic necrosis, systemic inflammation, and multi-organ dysfunction, in contrast to lean mice, where ethanol alone did not induce significant pancreatic changes (Yang et al., 2023). This model suggests that obese patients are particularly susceptible to progressing to severe AP and experiencing poor clinical outcomes. Our study corroborates this observation, underscoring the link between obesity and the severity of AP.

High BMI is intricately linked to the pathogenesis and etiology of AP, affecting its progression and prognosis, and it also correlates with genetic factors influencing AP occurrence. Recent discoveries highlight FGF21 as a key regulator of metabolic homeostasis with a potent anti-obesity effect (Lu, Li & Luo, 2021). Additionally, the loss of the FGF21 gene has been linked to increased intra-acinar triglyceride vacuole accumulation, and more severe edema and necrosis in chronic pancreatitis (Johnson et al., 2009). These findings emphasize the significance of genetic factors in the severity and evolution of pancreatitis, especially in relation to obesity. Moreover, while an association between BMI and AP recurrence is observed, not all high BMI individuals will experience relapses. Genetic factors might influence those with high BMI who have multiple AP recurrences in a short period.

Most studies indicate that a high BMI escalates the risk of acute pancreatitis, with no significant impact noted in cases of low BMI (Hansen et al., 2020). Contrary to these findings, our study reveals that not only obesity but also underweight status contributes to poor AP progression, suggesting that lower weight is not necessarily advantageous for AP patients. The underlying mechanisms and pathological changes warrant further investigation. While the course and severity of the disease are primarily influenced by inflammatory cells driving local and systemic immune responses, macro-level control can improve outcomes. In this context, BMI emerges as a valuable predictive and monitoring tool.

Conclusions

Our study indicates that a BMI exceeding 23.56 kg/m2 raises the risk of developing acute pancreatitis. Notably, a BMI above 25 kg/m2 but below 30 kg/m2 most significantly increases the risk of adverse disease progression, suggesting that severity does not necessarily escalate with higher BMI levels. BMI effectively predicts AP type; specifically, a BMI of 25.88 kg/m2 or higher suggests a potential progression to necrotizing pancreatitis. Furthermore, our findings link short-term (6 months) recurrence of AP with BMI, though this requires further validation due to the small sample size. Overall, BMI plays a crucial role in risk prediction and stratification for AP, underscoring the importance of monitoring patient BMI.

Supplemental Information

Supplemental Information 1 Clinical data of acute pancreatitis patients and control group participants.

The raw data shows high BMI is a risk factor for the occurrence and progression of AP and positively associated with recurrence, and underweight also exacerbates AP. The relationship between BMI and AP deserves continuous attention.

Click here for additional data file.

Additional Information and Declarations

Competing Interests

Author Contributions

Human Ethics

Data Availability

The authors declare that they have no competing interests.

Yuanzhen Bai conceived and designed the experiments, performed the experiments, analyzed the data, prepared figures and/or tables, authored or reviewed drafts of the article, and approved the final draft.

Guanwen Gong conceived and designed the experiments, authored or reviewed drafts of the article, and approved the final draft.

Reziya Aierken performed the experiments, prepared figures and/or tables, and approved the final draft.

Xingyu Liu performed the experiments, prepared figures and/or tables, and approved the final draft.

Wei Cheng analyzed the data, prepared figures and/or tables, and approved the final draft.

Junjie Guan analyzed the data, prepared figures and/or tables, and approved the final draft.

Zhiwei Jiang conceived and designed the experiments, authored or reviewed drafts of the article, and approved the final draft.

The following information was supplied relating to ethical approvals (i.e., approving body and any reference numbers):

Ethical approval for this study was granted by the ethics committee of Jiangsu Province Hospital of Chinese Medicine (No.2022NL-114-02). It was also conducted according to the China Good Clinical Practice in Research.

The following information was supplied regarding data availability:

The raw measurements are available in the Supplemental File.

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
