# Peer review of "A retrospective study investigating the clinical significance of body mass index in acute pancreatitis"

_PeerJ, doi:10.7717/peerj.16854_

## Round 0.1 · original submission · Major Revisions

Dear authors, thank you for your submission. At this stage, your work still requires significant revisions and answers to reviewers before being accepted. Please, refer to the reviewers' comments for further details.

**Language Note:** The review process has identified that the English language must be improved. PeerJ can provide language editing services - please contact us at copyediting@peerj.com for pricing (be sure to provide your manuscript number and title). Alternatively, you should make your own arrangements to improve the language quality and provide details in your response letter. – PeerJ Staff

·

Basic reporting

No comment

Experimental design

When BMI value is questioned as an obesity indicator, I'm not sure of the scientific interest of BMI in the setting of Acute Pancreatitis, Obesity is already considered a risk factor for acute pancreatitis and acute pancreatitis complications

Validity of the findings

The sample is small with insufficient power. Conclusions doubtful considering sample dimensions

Reviewer 2 ·

Basic reporting

Comment 1: The English in this study needs improvement. There are some terminologies that are not wrong but are not commonly used in epidemiological studies, as well as some expressions that may cause confusion or ambiguity.

For example: Lines 54-56: Is it a question or a statement?
Line 56: Change 'host factor' to 'risk factor'.
Line 64: What does 'it' refer to?
Lines 65-66: Please rephrase Aim 3.
Lines 74-75: This is confusing.
Line 76: What is meant by 'inducing factor'?
Line 67: The role of BMI in what?
Line 154: Change 'risk relationship' to 'relationship'.
Lines 151-152: Please format it."

Comment 2: Please kindly address the citation issue in line 54 of the introduction section. Numbers such as 80/20% need clear citation(s). In addition, while several meta-analyses have been conducted on this research question (attached below), only a limited number of independent studies have been discussed by the authors. Is BMI truly an underappreciated risk factor? Please elaborate on what additions will contribute to the existing evidence.

[1] Aune D, Mahamat-Saleh Y, Norat T, Riboli E. High Body Mass Index and Central Adiposity Is Associated with Increased Risk of Acute Pancreatitis: A Meta-Analysis. Dig Dis Sci. 2021;66(4):1249-1267. doi:10.1007/s10620-020-06275-6
[2] Premkumar R, Phillips AR, Petrov MS, Windsor JA. The clinical relevance of obesity in acute pancreatitis: targeted systematic reviews. Pancreatology. 2015;15(1):25-33. doi:10.1016/j.pan.2014.10.007
[3] Martínez J, Sánchez-Payá J, Palazón JM, Suazo-Barahona J, Robles-Díaz G, Pérez-Mateo M. Is obesity a risk factor in acute pancreatitis? A meta-analysis. Pancreatology. 2004;4(1):42-48. doi:10.1159/000077025

Experimental design

Comment 1: In lines 63 to 68, the authors listed all the study goals, covering questions about descriptive analysis (aim 3), association (aims 4 and 5), causation (aim 2), and prediction (aim 1). From my personal standpoint, this study seems to be focusing on the association between BMI and the occurrence of AP, along with a sensitivity analysis investigating the association among different subgroups, such as patients with different disease types and various BMI groups. I suggest identifying the primary research goal for this study and ensuring that the primary analysis is well-conducted to generate reliable evidence for the research question. Additionally, rephrase the goals and organize them.

Comment 2: The methods were not adequately described for reproducibility. The authors need to justify why only two years of data were included instead of extending the study period for a larger sample size, given that this is a retrospective analysis. Details about model development, such as how the regression and prediction models were constructed, need to be presented.

Other suggestions are as follows:
1. Be more specific about the covariates, such as defining 'inducing factor' and clarifying what 'initial/recurrence' refers to. Clarify whether health behavioral information is combined (e.g., cigarette/alcohol habits) and specify the units of variables (e.g., hospital stay in days or months).
2. The selection of the control group is not clear enough. For instance, were those 85 patients selected from the same specialty or from other specialties? It is important to ensure that patients in the disease group do not have a higher risk of disease occurrence and progression compared to the control group.
3. When is the severity of acute pancreatitis evaluated? Given the inclusion of post-admission factors, it is crucial to understand this for the proper utilization of predictors.
4. Line 83 requires a citation for China Good Clinical Practice in Research
5. Discuss how the exclusion criteria will affect the generalizability of the outcomes.
6. Describe the methods used to handle missing data.
7. Describe the process of model diagnostics and how the model fit is evaluated.

Validity of the findings

Comment 1: The novelty and limitations of the study were not sufficiently addressed.

Comment 2: The results may have been overinterpreted in the discussion and conclusion sections. For example:
1. Line 241-242: Avoid interpreting the association as a causal relationship; avoid using words such as "lead to".
2. This study did not conduct any external validation; therefore, the suggestion of using BMI at admission to predict the occurrence or type of disease is worth reassessing.
3. The discussion of the relationship between BMI and the outcome should differentiate between the research frameworks for association, causation, and prediction. For example, BMI is a risk factor/cause/predictor of the outcome of interest.

Comment 3: Please find below a few suggestions on the result section:

Line 154: It would be helpful to present the results with specific numbers to minimize any ambiguity.
Line 156: Could you kindly clarify the odds ratios of which subgroups you are referring to?
Line 167-169: It may be beneficial to consider the potential impact of the sample size when interpreting the results.
Line 173-174: It appears that using BMI to predict the incidence might be better described as estimating the risk of the occurrence.

·

Basic reporting

The authors have provided a significant concern of AP patients and the potential risk factors. According to the detailed analysis and explanation, it’s convincing to summarize that BMI can be added as a risk factor to increase the diagnostic accuracy of AP.

Experimental design

The authors did a comprehensive analysis of the potential risk factors that may related to AP. According to figure 2, it has been shown that both low and high BMI are strongly associated with AP.

Validity of the findings

ROC curve in figure 3 confirmed the hypothesis.
Conclusions are well stated, and the discussion linked the original research question and the supporting results.

Additional comments

Page 6, line 48-49, it would be proper to say “one of the leading causes…”
Line 54, reference not found.

---

## Round 0.2 · accepted · Accept

Many thanks for your throughout work on the revised version. I can now suggest your manuscript for publication.

·

Basic reporting

No comment

Experimental design

Research within aims and scope of the Journal
Research question is well defined, the study explores the well-known association between obesity and acute pancreatitis, (and of malnutrition and risk) providing more information. Although BMI is being questioned as indicator for fat risk association it still is easier to obtain than most alternatives
Investigation was performed with adequate technical and ethical standards.
Methods are described with sufficient detail & information to replicate.

Validity of the findings

Results contribute to consolidate and tune the information regarding the relation of Acute Pancreatitis and BMI, both in occurrence and in outcome. Raises questions regarding the relation of normal BMI with more severe forms of pancreatitis
Particularly consider figure 1 very expressive
Conclusion are aligned with the results and contributes to consolidate previous information,